# Combination of Biological Therapy in Severe Asthma: Where We Are?

**DOI:** 10.3390/jpm13111594

**Published:** 2023-11-10

**Authors:** Lorenzo Carriera, Marta Fantò, Alessia Martini, Alice D’Abramo, Genesio Puzio, Marco Umberto Scaramozzino, Angelo Coppola

**Affiliations:** 1Facoltà di Medicina e Chirurgia, Università Cattolica del Sacro Cuore, 00168 Rome, Italy; alessia.martini@guest.policlinicogemelli.it; 2UOSD Allergologia e Immunologia Clinica, A.O. San Donato, USL Toscana Sud Est, 52100 Arezzo, Italy; marta.fanto@uslsudest.toscana.it; 3UOC Pneumologia, Ospedale San Filippo Neri-ASL Roma 1, 00135 Rome, Italy; alice.dabramo@aslroma1.it (A.D.); genesio.puzio@hotmail.it (G.P.); angelo.coppola@aslroma1.it (A.C.); 4Ambulatorio “La Madonnina”, Reggio Calabria/Villa Aurora Hospital, 89100 Reggio Calabria, Italy; scaramozzinomarco91@gmail.com; 5UniCamillus, Saint Camillus International University of Health Sciences, 00131 Rome, Italy

**Keywords:** severe asthma, type 2 comorbidities, biologics, dual biologic therapy, combination biologic therapy

## Abstract

Biological drugs have revolutionized the management of severe asthma. However, a variable number of patients remain uncontrolled or only partially controlled even after the appropriate administration of a biologic agent. The combination of two biologics may target different inflammatory pathways, and it has been used in patients suffering from uncontrolled severe asthma with evidence of both allergic and eosinophilic phenotypes or severe asthma and type2 comorbidities. Combination therapy has also been used to handle anti-IL4/13R induced hypereosinophilia. There is insufficient data on combining biologics for the treatment of severe uncontrolled asthma and type 2 comorbidities, also because of the high cost, and currently no guideline recommends dual biologic therapy. A systematic search was performed using the Medline and Scopus databases. Published data on concurrent administration of two biological drugs in severe, uncontrolled asthma patients has been reported in 28 real-world studies and 1 clinical trial. Data extraction was followed by a descriptive and narrative synthesis of the findings. Future studies should be conducted to further assess the safety, efficacy, and cost-effectiveness of this therapeutic strategy.

## 1. Introduction

Asthma is a chronic inflammatory disease of the airways. It affects approximately 300 million people worldwide, and about 5–10% of them suffer from severe asthma, defined as “asthma which requires treatment with high dose inhaled corticosteroids (ICS) plus a second controller (and/or systemic corticosteroids) to prevent it from becoming ‘uncontrolled’ or which remains ‘uncontrolled’ despite this therapy” [1].

Asthma is heterogeneous in terms of the natural history of the disease, clinical presentation, severity, and response to treatment, and this heterogeneity reflects the underlying distinct pathophysiologic mechanisms (asthma endotypes) [2]. The main pathogenic mechanisms are often categorized as T2-high and T2-low.

T2-high endotypes have an immune–inflammatory response driven by T helper type 2 (Th2) cells and type 2 innate lymphoid cells (ILC2) and mediated by type 2 cytokines, such as interleukin (IL)-4, IL-5, and IL-13, and include the allergic and non-allergic eosinophilic phenotypes [3]. Biomarkers, such as blood eosinophil count (BEC), total and specific IgE, and fractional exhaled nitric oxide (FeNO), can be used as indicators of type 2 asthma endotypes [4]. T2-low asthma is defined by the absence of T2-high biomarkers and is either driven by neutrophilic (dependent on a Th1/Th17-mediated immune response) or paucigranulocytic inflammation [5].

Biological drugs have revolutionized the management of severe asthma by significantly reducing exacerbations and the need for systemic corticosteroids, improving lung function, and improving patient quality of life [6]. However, a variable number of patients with severe asthma remain poorly controlled and experience function decline, despite the use of maximal inhaled therapy, systemic corticosteroids, and the addition of biologic therapy as well. Many are the possible reasons for a suboptimal response [7,8].

Deciding to stop or switch biological therapy in these patients may be challenging, especially when they qualify for more than one biologic because they present clinical characteristics that overlap two endotypes. In fact, during the lifetime, environmental stimuli have demonstrated the ability to induce modifications in the immunological and inflammatory profiles of asthmatic patients, leading to a complex overlap of phenotypes and endotypes [9].

Asthmatic patients frequently suffer from type 2 comorbidities [10]. Type 2 inflammation is implicated in several respiratory and other diseases, such as atopic dermatitis (AD), urticaria, allergic rhinitis (AR), chronic rhinosinusitis with nasal polyposis (CRSwNP), and eosinophilic esophagitis (EoE) [10,11]. Type 2 immunity diseases share some pathophysiological characteristics. Epithelial dysfunction is a common trait across type 2-driven diseases, and even if they develop across different barriers, the inflammatory cascade follows some common paths. For example, eosinophils are key effectors involved in type 2 immunity. Their role is well established in asthma and CRSwNP, but also in AD, where they are classically elevated in patients’ serum and infiltrate lesional skin [12]. There are clear associations between type 2-driven diseases. They can be successive in time or frequently coexist [13], especially considering asthma and CRSwNP, which led to the concept of unified airway disease [14].

The presence of these comorbidities complicates therapeutic management [15]. For example, published data in the literature show that the presence of CRS is associated with worse outcomes and an increased risk of exacerbations in patients with asthma [14]. Since the management of patients with type 2 inflammatory conditions is complex, a multidisciplinary approach involving a pneumologist, allergologist, ear-nose-throat (ENT) specialist, and dermatologist is considered necessary to optimize care [10].

All currently available asthma biologics target cytokines or cells within the T2 inflammatory pathway. These drugs interfere with T2-inflammation in different ways: omalizumab targets IgE; mepolizumab and reslizumab target IL5; benralizumab targets the IL5 receptor; and dupilumab targets IL4 and IL13. There is no specific therapy for patients with T2-low inflammation.

In this scenario, tezepelumab, recently approved as an add-on maintenance treatment for severe asthma [16,17], represents a big innovation and is a potential “game changer “in the management of severe and difficult-to-treat asthma. Tezepelumab is a fully human monoclonal antibody that targets thymic stromal lymphopoietin (TSLP), an epithelial-derived cytokine released in response to multiple triggers, preventing its interaction with the receptor and thus inhibiting multiple downstream inflammatory pathways [6]. By blocking TSLP upstream of the inflammatory cascade, tezepelumab managed to reduce asthma exacerbations in patients with either high or low levels of T2 biomarkers [16,17]. It was able to down-regulate BEC, FeNO, and serum total IgE levels [18].

Before tezepelumab became available worldwide, the combination of biologic therapy represented an option reported in some real-world experiences for asthmatic patients who remained uncontrolled with one biologic drug, in particular those with evidence of both allergic and eosinophilic inflammation, and has been used in patients with severe asthma who suffered from type 2 comorbidities. The combination of two biologics may target different biological pathways, providing additional improvement. Dual biologic therapy has also been employed in a few cases for the management of anti-IL4/13R-induced hypereosinophilia and eosinophils-related disorders. Currently, no guideline recommends concurrent administration of two biologics for the treatment of asthma and T2 comorbidities.

Limited preliminary data, mostly derived from observational studies and case series, suggests this strategy may be safe in selected patients even if combining therapies is not permitted in some countries or by certain payors [19]. Before considering the opportunity to combine biological drugs, a complete reassessment of the patient at the time of treatment failure or partial or deterioration of response can be recommended, which may include pulmonary function tests, airway inflammatory cell count, imaging and/or bronchoscopy for their complicating disorders, and detecting neutralizing drug antibodies, reported in 1–4% of participants in clinical trials [20].

The aim of this narrative review is to summarize the current evidence deriving from published data about the efficacy and safety of combining asthma-approved biologics, either to achieve better control of uncontrolled severe asthma or severe asthma and uncontrolled T2 comorbidities, and provide an overview of this published data regarding the conditions for which a combination of biologic agents was administered.

## 2. Materials and Methods

### 2.1. Search Strategy

A systematic search was performed following the Preferred Reporting Items for Systematic Reviews and Meta-Analyses (PRISMA) guidelines to identify the current literature about dual therapy with asthma-approved biologics for a combined treatment of severe asthma and for treating concurrent severe asthma and type 2 comorbidities. The search was conducted on the online databases MEDLINE (PubMed) from the National Library of Medicine (NLM) and Scopus, from inception to September 2023, and was followed by manual literature searches in the reference lists of the included articles to identify additional articles about this topic. The research string was as follows: (dual [Title] OR combination [Title] OR simultaneous [Title] OR combining [Title] OR combined [Title] OR concomitant [Title]) AND (biologic [Title] OR biologics [Title] OR biologic therapy [Title] OR therapy [Title] OR biologics therapy [Title] OR monoclonal antibody [Title] OR monoclonal antibodies [Title] OR biologic treatment [Title] OR targeting [Title]) AND (severe asthma [Title] OR uncontrolled asthma [Title] OR t2 comorbidities [Title] OR type 2 comorbidities [Title] OR asthma and CRSwNP [Title] OR asthma and atopic dermatitis [Title] OR asthma and urticaria [Title] OR asthma and EGPA [Title] OR asthma and eosinophilic esophagitis [Title] OR asthma and ABPA [Title] OR omalizumab [Title] OR IgE [Title] OR anti-IgE [Title] OR mepolizumab [Title] OR reslizumab [Title] OR IL5 [Title] OR anti-IL5 [Title] OR benralizumab [Title] OR IL5R [Title] OR anti-IL5R [Title] OR dupilumab [Title] OR IL4/IL13 [Title] OR anti-IL4/IL13 [Title] OR Tezepelumab [Title] OR TSLP [Title] OR anti-TSLP [Title] OR omalizumab and mepolizumab [Title] OR omalizumab and benralizumab [Title] OR IgE and IL5 [Title] OR omalizumab and dupilumab [Title] OR mepolizumab and dupilumab [Title] OR benralizumab and dupilumab [Title] OR omalizumab and reslizumab [Title] OR reslizumab and dupilumab [Title] OR IgE and IL4/IL13 [Title] OR IL5 and IL4/IL13 [Title]).

### 2.2. Study Selection

Clinical trials, observational studies, case series, case reports, and letters were included, with no time limitation and in any language. Studies with both adult and pediatric patients have been considered. Studies available only as abstracts were equally included in this analysis. The exclusion criteria were: any publication not focusing on this topic; reviews; and studies about combining asthma-approved biologic agents with other monoclonal antibodies for different therapeutic indications. Non-relevant studies were excluded by title. The eligibility of studies was assessed using the inclusion criteria, and studies that did not fit were excluded. Because the goal of the review was to analyze all published data about the clinical efficacy and safety of dual biologic therapy for asthma and type 2 comorbidities, and many of the included studies showed heterogeneity and were conducted on small sample sizes, a quality assessment was not performed.

### 2.3. Data Extraction

Data extraction was conducted by one author (L.C.) to be subsequently discussed and checked by a second author (A.C.). The extracted data included the following items: author/year, study design, population size, pathologies for which dual biologic therapy was administered, prescribed biologic agents, follow-up, efficacy, and safety.

### 2.4. Data Synthesis

We descriptively summarized the data and undertook a narrative and critical synthesis of the information, reporting the quantitative findings of individual studies whenever provided by the authors.

## 3. Results

The initial literature search generated 137 potentially eligible articles from the aforementioned databases, plus 11 records identified by manual search. A total of 60 duplicates were identified and removed. After excluding 59 articles (51 off-topic, 1 review, and 7 about combining asthma-approved biologic agents with other monoclonal antibodies for different therapeutic indications), only 29 articles were included in this review according to the prespecified inclusion and exclusion criteria. A flow chart showing the study selection is presented in Figure 1. The data derive from 28 real-world studies and 1 clinical trial. A total of 27 studies took into consideration adult patients, while 2 case reports described pediatric patients. Table 1A–D summarizes the studies included in this review. The characteristics of the studies in which two biologics were administered for the treatment of uncontrolled severe asthma are presented in Table 1A. Two studies in which cycling biologic therapy was used are summarized in Table 1B. The cases in which dual therapy was given for severe asthma and type 2 comorbidities are presented in Table 1C,D. This illustrates the characteristics of 4 studies in which a dual biologic therapy was administered to deal with dupilumab-induced hypereosinophilia and eosinophils-related disorders, including cases of eosinophilic granulomatosis with polyangiitis (EGPA).

### 3.1. Dual Biologic Therapy for Severe Asthma Insufficiently Controlled with a Biologic Monotherapy

A recent placebo-controlled randomized phase II study [21] evaluated a new anti-IL33 monoclonal antibody, itepekimab, and dupilumab, in combination or alone. It involved a total of 296 patients. Itepekimab treatment improved asthma control and quality of life, as compared with placebo, and improved lung function. Anyway, no beneficial effect was observed compared to placebo when itepekimab and dupilumab were combined. In terms of adverse events, the combination therapy was similar to the placebo group.

Serajeddini et al. [22] illustrated in a case series the administration of dual biologic therapy to eight patients with severe asthma. The authors reported that despite partial improvement after the administration of the first biologic agent, 4 of the 8 patients could not stop taking oral corticosteroids and remained OCS-dependent, 7 patients continued to experience 2 or more moderate to severe exacerbations per year, and 6 patients continued to experience severe sinonasal symptoms. In this study, 4 patients received a combination of benralizumab and dupilumab, 2 patients received a combination of mepolizumab and dupilumab, and the other 2 patients received reslizumab plus dupilumab. Dupilumab was added when there were clinical signs of uncontrolled IL4/IL13 inflammation, driving asthma pathology even under an anti-IL5 monoclonal antibody (mAB): elevated FeNO, airway smooth muscle dysfunction, and severe airway hyperresponsiveness (AHR) with significant postbronchodilator reversibility (pBDR). In all cases, a significant clinical and functional improvement was observed with the concurrent treatment with two biologics, along with a reduction of inflammatory biomarkers. Sputum IL4, IL13, and IL5 levels were measured, with a >90% reduction after the addition of dupilumab as a second biologic agent. Anyway, two patients failed to obtain complete symptoms control.

Thomes and Darveaux [23] reported the cases of three patients who started a dual biologic therapy due to a lack of control over monotherapy with omalizumab. Since mepolizumab was added, the combination with a second biologic agent allowed to reduce the number of exacerbations, taper the daily OCS dose, and provide better control of symptoms.

An Italian study by Baccelli et al. [24] reported the case of a 68-year-old woman with allergic and eosinophilic corticosteroid-dependent severe asthma. She was initially prescribed omalizumab, which improved her symptoms and increased her quality of life. Anyway, due to frequent exacerbations and worsening of symptoms, after eleven years of biologic therapy, mepolizumab was then added as a second biologic agent since the patient refused to stop anti-IgE therapy. The combination therapy reduced the number of exacerbations without any new hospitalization, improved the quality of life, increased the exercise capacity measured with the 6MWT, and improved respiratory function. The daily OCS was slightly tapered but not interrupted. At 3-year follow-up, combination therapy proved to be safe, and no adverse events were observed since the second biologic was introduced, the authors report.

Dedaj and Unsel [25] described the use of the combination of omalizumab and mepolizumab in a 55-year-old woman with severe eosinophilic asthma and elevated IgE levels, with no controlled symptoms despite maximal inhaler therapy plus other controllers (montelukast and azithromycin) and ongoing biologic therapy with omalizumab. Maintenance OCS therapy was started due to frequent exacerbations. Since mepolizumab was added, the authors report that the patient consistently reduced the daily dose of the OCS, had fewer exacerbations, and did not have any ER visits or hospitalizations. No side effects were described.

Bergmann et al. [26] presented the case of a 53-year-old man with severe, uncontrolled allergic and eosinophilic asthma. The patient was already receiving maximal therapy with high doses of ICS and LABA, tiotropium, montelukast, and oral steroids, and was prescribed long-term oxygen therapy. Comorbidities included nasal polyposis, aspirin-exacerbated respiratory disease (AERD), and severe bronchiectasis, predominantly in both lower lung lobes. The respiratory function was significantly compromised. The patient was a candidate for a lung transplant. Omalizumab was started and led to a rapid improvement. After 18 months, anti-IL5 mepolizumab was added to omalizumab because of the increased blood eosinophil count and worsening of asthma symptoms. Undergoing dual biological therapy, symptoms rapidly reduced, and lung function improved even more than with omalizumab alone. Mepolizumab was later withdrawn in order to assess the necessity of the second biologic drug, but the discontinuation of mepolizumab determined a loss of asthma control, deterioration of respiratory function, and eosinophilia, which increased again. Therefore, benralizumab was started (administered 2 weeks after the omalizumab injections), resulting in an immediate improvement of respiratory function and asthma symptoms, and the patient did not need a transplant anymore because, as a follow-up HRCT scan showed, dual biologic therapy reduced bronchial wall thickening and mucous plugging, the authors report. The patient received omalizumab and mepolizumab for 17 months and omalizumab and benralizumab for at least 7 months. No side effects were described.

Domingo et al. [27] described the case of a 55-year-old woman with corticosteroid-dependent allergic asthma, initially treated with omalizumab. Despite a very limited improvement, the patient continued omalizumab for ten years until mepolizumab became available and a switch could be possible since the patient had a high BEC. Anyway, the clinical improvement was poor, also because of a new flare-up of allergy symptoms, and a combination therapy with omalizumab and mepolizumab was then performed. Without any side effects, an improvement in the respiratory function could be observed, and the OCS dose was tapered.

A recent case report in Turkey [28] illustrated the combination of mepolizumab and omalizumab in a 52-year-old patient with allergic asthma, uncontrolled with either omalizumab or mepolizumab treatment alone. The concurrent administration of two biologics determined a significant improvement in the control of symptoms, improved quality of life, and increased respiratory function. OCS was stopped. No side effects were observed.

Fox and Rotolo [29] described the case of a 12-year-old female patient who received two biologic agents, omalizumab and mepolizumab, to control her severe asthma. Despite maximal therapy, she had had multiple exacerbations and hospital admissions, so omalizumab was started with an initial response. One year later, mepolizumab was added due to frequent exacerbations and an elevated blood eosinophil count. The patient received both biologics for 24 months with improvement in the control of asthma symptoms and without remarkable side effects (only a mild headache was reported by the authors). Since starting mepolizumab, she has not required any oral corticosteroids in the follow-up period.

Phan et al. [30] described the case of a 16-year-old female with severe allergic asthma and allergic rhinitis treated with the concurrent administration of omalizumab and mepolizumab. The patient was already on omalizumab, and mepolizumab was later added because of peripheral hypereosinophilia and OCS dependence. Since the start of dual biologic therapy, symptoms have improved, and the patient has had a higher quality of life.

### 3.2. Cycling Biologic Therapy for Severe Asthma

Two cases of cycling biologic therapy have been described. The former [31] is the case of a 43-year-old woman with severe eosinophilic asthma, eosinophilic chronic rhinosinusitis (ECRS), and eosinophilic otitis media (EOM), receiving cycling therapy with dupilumab and benralizumab (a cycle of dupilumab therapy, four times every 2 weeks, a month after a single administration of benralizumab). The latter [32] describes the case of a 47-year-old man with severe allergic and eosinophilic asthma and chronic rhinosinusitis with nasal polyposis. The cycling biologic therapy administered for this patient comprised dupilumab every 2 weeks and a mepolizumab injection the following month. Since cycling therapy was started, the patient did not experience any exacerbations requiring OCS, emergency department visits, or hospital admissions, and the control of asthma symptoms considerably improved, the authors report.

### 3.3. Dual Biologic Therapy for Severe Asthma and Type 2 Comorbidities

Pitlick and Pongdee [33] described 25 patients who received and safely tolerated two biologic agents at the same time, between 2015 and 2021. A total of 10 patients were treated with a dual biological therapy to obtain asthma control (4 received a combination of omalizumab and benralizumab, while 6 patients received omalizumab plus mepolizumab). A total of 3 other patients had asthma and other comorbid conditions: 1 patient with CRSwNP received mepolizumab and dupilumab, 1 patient with EGPA received omalizumab and mepolizumab, and another patient with CSU was treated with omalizumab and mepolizumab. A total of 12 patients received two biologics for the treatment of type 2 comorbidities, 6 for the same condition, and 6 for separate conditions. No episode of anaphylaxis or other allergic reaction was reported among patients receiving a combination of biologics; there was no hepatic or renal impairment, pneumonia or infectious complications, immune system dysfunction, or new cases of cancer. The efficacy of dual biologic therapy was not described in this study.

A case series by Lommatzsch et al. [34] described 25 patients treated with several biologic combinations. 15 patients concomitantly received 2 biologics approved for severe asthma: 7 received a dual treatment to obtain asthma control, and 8 were treated with another asthma biologic for the management of severe asthma and T2-high comorbidities. 2 patients with AD were treated with either benralizumab or mepolizumab and dupilumab; 1 patient with CSU received benralizumab and omalizumab; 1 patient with EGPA received omalizumab and mepolizumab; 4 patients received either benralizumab or mepolizumab and dupilumab for CRSwNP. The start of the dual therapy was preceded by an attempt to switch to another biologic as a monotherapy. There were no adverse effects reported, but the dual treatment was stopped in 4 patients, all in the uncontrolled asthma group, because of clinical ineffectiveness. The other 10 patients in the case series received one biologic for asthma plus another for the treatment of a concomitant but different disease, so they were not considered in this analysis.

Otten et al. [35] evaluated 94 patients who switched from one biologic to another for the treatment of CRSwNP and asthma because they could not achieve good control of their upper and/or lower airway disease with the first prescribed biological drug or experienced significant side effects. A total of 7 patients received in the study a combination of two biologics. A total of 4 patients received a combination of dupilumab and benralizumab, 2 patients received dupilumab and mepolizumab (later, 1 patient switched back to dupilumab only after less than a year of double treatment), and 1 patient received reslizumab and dupilumab. The reasons for dual therapy were the following: 3 patients had persistent hypereosinophilia during dupilumab, 2 patients developed hypereosinophilic syndrome (HES) when treated with dupilumab but had insufficient control of CRSwNP when treated with anti-IL5 only, 1 patient had insufficient control of asthma on dupilumab only and insufficient control of CRSwNP when on anti-IL5 treatment only, and 1 patient developed a side effect (arthritis) possibly related to dupilumab but had insufficient control of CRSwNP on anti-IL5 only after dupilumab discontinuation. The study also described more in detail the case of a patient who received dual biological therapy, a 57-year-old woman with OCS-dependent severe asthma, CRSwNP, and EOM. She started biologic therapy with mepolizumab, and asthma symptoms significantly improved. However, CRSwNP and EOM symptoms did not improve, so she switched to reslizumab. Later, she returned to the hospital complaining of severe nasal symptoms, no sense of smell, impaired hearing, and otitis media with effusion. Therefore, the patient started therapy with dupilumab. She experienced a rapid and significant improvement in both upper (nasal polyps totally disappeared) and lower airway symptoms. Anyway, it was not possible to reduce the OCS, and because of blood hypereosinophilia and progressively increasing respiratory symptoms, it was decided to switch the patient to benralizumab. With benralizumab, asthma improved, but nasal polyps and EOM (significantly impairing her hearing) relapsed. It was finally decided to combine dupilumab and benralizumab. The combination of two biologics allowed for good control of both upper and lower airways, and OCS was discontinued.

Ortega et al. [36] discussed the cases of 3 patients, all with eosinophilic and allergic severe asthma and type 2 comorbidities, treated with multiple asthma biologic drugs concurrently. A 61-year-old woman with AD, allergic rhinitis, and ABPA received omalizumab and dupilumab. A 60-year-old woman with CSU, allergic rhinitis, and CRSwNP and a 43-year-old man with CRSwNP, ABPA, and allergic conjunctivitis both received a combination of omalizumab and mepolizumab/benralizumab. All patients experienced an improvement in the control of the disease and were able to stop OCS without any adverse effects that could be attributed to biologics, but Patient 3 was hospitalized while on omalizumab/benralizumab because of an asthma exacerbation and was switched to dupilumab monotherapy. Notably, two of the patients in this study experienced worsening of allergic symptoms after discontinuing omalizumab. When the anti-IgE biologic was reintroduced, patients regained better symptom control, the authors report.

Caskey and Kaufman [37] reported the case of a 51-year-old man with severe eosinophilic asthma. His condition was complicated by CRSwNP, AERD, and CSU. Initiation of therapy with mepolizumab led to an improvement in the control of asthma symptoms. However, the patient continued to have frequent exacerbations of chronic idiopathic urticaria, resistant to common treatment with high doses of H1-antihistamines. Although asthma exacerbations were controlled in this patient thanks to biologic therapy with mepolizumab, control for symptoms related to CSU was not achieved. Omalizumab was therefore added to separately treat urticaria, and the administration of a second biologic drug successfully reduced urticaria symptoms.

Volpato et al. [38] described the case of a 60-year-old male patient with severe asthma and comorbid severe chronic rhinosinusitis with nasal polyposis, AERD, and CSU. Given perennial sensitizations, elevated serum total IgE levels, and several exacerbations necessitating high doses of OCS, omalizumab was started. The number of exacerbations was reduced during the first years of treatment with omalizumab, asthma control improved, and the daily OCS dose was tapered but could not be completely discontinued. After an initial good response to omalizumab, BEC increased, and the patient experienced new severe exacerbations. Omalizumab was suspended, and mepolizumab was introduced. Asthma control improved, but the patient experienced frequent episodes of urticaria, despite appropriate therapy, and was admitted to the emergency department suffering from angioedema and thus necessitating OCS therapy again. Therefore, it was decided to reintroduce omalizumab as a concomitant biological therapy for chronic idiopathic urticaria. The patient received monthly injections of mepolizumab and omalizumab on the same day on different shoulders. The combination of omalizumab for the treatment of urticaria and anaphylactic manifestations and mepolizumab for the treatment of eosinophilic asthma and chronic rhinosinusitis successfully allowed to control symptoms and improve the quality of life of this patient.

A case report [39] described the case of a 50-year-old female who was receiving omalizumab for the treatment of her severe asthma. She had experienced frequent exacerbations requiring systemic corticosteroids and continued to suffer from CRSwNP. Due to her uncontrolled asthma and peripheral hypereosinophilia, the patient was switched to benralizumab, and omalizumab was discontinued. Her asthma and rhinosinusitis symptoms considerably improved, but after a week she developed chronic urticaria, refractory to high-dose antihistamines, thought to be related to omalizumab discontinuation. The start of a dual biologic therapy finally improved control of both urticaria and asthma symptoms, reduced the need for OCS use, and no adverse events were reported.

Recently, a case report [40] described how a patient with severe asthma and urticaria, both uncontrolled with biologic monotherapy, was successfully treated with a combination of omalizumab and mepolizumab. No adverse events were observed after 6 months of follow-up.

Tongchinsub and Carr [41] presented the case of a 56-year-old female patient with severe eosinophilic asthma, chronic rhinosinusitis with nasal polyps, and eosinophilic mastoiditis. She was treated with omalizumab for multiple years. However, she still required oral corticosteroids, so mepolizumab was added due to only partial improvement in asthma and sinus symptoms and persistent hypereosinophilia. After dual biologic therapy was commenced, her respiratory symptoms and the control of both asthma and sinus mastoiditis rapidly improved, and the patient stopped assuming OCS, the authors report. The patient did not experience any side events.

A case report [42] described the successful concurrent administration of mepolizumab and omalizumab for the treatment of a 67-year-old female diagnosed with severe asthma, eosinophilic esophagitis, and gastroenteritis. To relieve both asthma and GI symptoms (intractable nausea, vomiting, and diarrhea), OCS and inhaled corticosteroids were prescribed. Prednisone’s daily dose could not be tapered due to persistent asthma and GI symptoms, and BEC remained high despite administered therapies. As eosinophils were thought to be driving both asthma and the GI inflammation, she started mepolizumab. Eosinophilic gastroenteritis was effectively controlled with mepolizumab. However, her asthma symptoms were still present. Due to recurrent asthma exacerbations in the fall, thought to be related to ragweed, therefore suggesting an allergic drive, omalizumab was added. Thanks to the concurrent administration, respiratory symptoms improved too, and OCS was stopped. In this patient, mepolizumab successfully treated the visceral GI eosinophilia while omalizumab treated the allergic asthma, the authors report. No adverse events were reported.

Laorden et al. [43] presented a group of 3 patients with severe asthma and allergic bronchopulmonary aspergillosis (ABPA) who improved upon being prescribed omalizumab but, after a few years, experienced worsening of symptoms and loss of asthma control. Since an eosinophil-driven inflammatory pathway was considered responsible for the clinical deterioration, anti-IL5/IL5R therapy was added. Omalizumab was not discontinued because it had allowed it to control symptoms, prevent exacerbations, and taper OCS for a long period of time. The combination of omalizumab and either mepolizumab or benralizumab made it possible to achieve significant control of symptoms, reduce exacerbations, improve respiratory function, and taper OCS, the authors report. Compared to other studies, in this one the dose of omalizumab was progressively stepped down, therefore reducing costs deriving from a dual biologic therapy.

Curtiss et al. [44] described the cases of two patients with refractory OCS-dependent ABPA who experienced adverse effects because of chronic OCS therapy, failed single biologic therapy, and successfully responded to dual biologic therapy (IL5 and IL4/IL13).

Altman et al. [45] reported the case of a 58-year-old woman diagnosed with ABPA. Despite receiving high doses of OCS and antifungal therapies, the patient showed no clinical improvement. Omalizumab was added, with a steroid-sparing effect and fewer ABPA exacerbations. Although omalizumab therapy halted the patient’s clinical decline, only the addiction to a second biologic agent (mepolizumab) resulted in complete control of symptoms and led to the full return of ADLs and the discontinuation of OCS and supplemental oxygen.

### 3.4. Dual Biologic Therapy for Reactive Hypereosinophilia and Eosinophils Related Disorders

A case report [46] described eosinophilic vasculitis as a complication of hypereosinophilia induced by dupilumab in a patient with severe eosinophilic non-allergic asthma and chronic rhinosinusitis. Dupilumab improved asthma symptoms and allowed the withdrawal of OCS. However, BEC progressively increased, and life-threatening eosinophilic vasculitis-like manifestations appeared. In this case, dupilumab discontinuation was initially considered, but, in order to avoid resuming steroids and since previous anti-IL-5 mepolizumab monotherapy had failed to control asthma symptoms, it was chosen not to stop dupilumab and to add-on benralizumab. The combination therapy induced eosinophils depletion and complete resolution of vasculitis symptoms, the authors report.

Another study [47] reported the case of a 42-year-old man with severe eosinophilic asthma, in which EGPA became active immediately after switching from benralizumab to dupilumab. The patient had previously received mepolizumab and then benralizumab. However, asthma symptoms could not be controlled, and the OCS dose could not be tapered. Therefore, the patient was switched to dupilumab. After a few weeks, he was admitted to the hospital because of an acute worsening of symptoms, presenting parenchymal opacifications in both lungs on a chest CT, and was finally diagnosed with EGPA. Dupilumab was discontinued, and EGPA treatment with mepolizumab 300 mg, OCS, and cyclophosphamide was started. After the treatment for EGPA was initiated, however, a loss of control of asthma symptoms was observed, respiratory function worsened, and FeNO levels markedly increased. For these reasons, it was decided to prescribe dupilumab as a concomitant treatment, and after it was administered, both symptoms and FEV1 rapidly improved. According to the authors, the concomitant use of mepolizumab and dupilumab was safe and effective to control the disease, and it was continued for more than 16 weeks without any adverse events reported.

Philipenko et al. [48] described how the concurrent administration of mepolizumab and dupilumab could alleviate dupilumab-induced conjunctivitis, providing additional improvement in asthma control. A 28-year-old male with severe atopic dermatitis and severe eosinophilic asthma started biological therapy with dupilumab. After four weeks, he developed bilateral conjunctivitis and hypereosinophilia, so mepolizumab was initiated. Conjunctivitis resolved, and asthma symptoms improved even more than with dupilumab alone. The patient remained on combination therapy without any new asthma exacerbations, and dupilumab withdrawal was not necessary.

Briegel et al. [49] described the case of a 24-year-old woman with a probable diagnosis of ANCA-negative EGPA, treated with azathioprine and prednisolone. Anti-IL5 treatment was started with steroid sparing intent (first mepolizumab, then switched to benralizumab). The therapy led to an improvement in asthma symptoms and allowed for the tapering of the dose of prednisolone. Because of a severe recurrence of nasal polyposis and a history of three prior operations, treatment was switched to dupilumab rather than letting the patient undergo another operation. Symptoms related to nasal polyposis initially improved, but after 3 months of therapy with dupilumab, pulmonary symptoms worsened and a new-onset dry cough appeared. Blood eosinophils increased despite ongoing therapy with prednisolone and azathioprine, so the treatment was switched back to benralizumab. After nasal polyposis recurred again, dual treatment with dupilumab and benralizumab was given, resulting in a consistent improvement of both symptoms related to asthma and nasal polyposis. The combination of biologics allowed for a stable reduction in prednisolone dose, and no side effects related to the dual therapy were observed, the authors report.

## 4. Discussion

The simultaneous administration of omalizumab and an anti-IL5 biologic agent was the most commonly used combination in real-life studies for the treatment of severe, uncontrolled asthma. The main reasons reported by the authors of these studies for this strategy were an increase in blood eosinophil count, worsening of asthma symptoms, and OCS dependence on omalizumab therapy [23,24,25,26,28,29,30]. However, no data are provided in some studies regarding the reasons that led to starting a combination therapy instead of switching to anti-IL5 or how long it has been administered. The OSMO study [50] showed that patients with asthma not optimally controlled on omalizumab but eligible for both omalizumab and mepolizumab therapy, after switching to mepolizumab, had a reduction in asthma exacerbations and a clinically significant improvement in asthma control and lung function. The study also wanted to point out if a direct switch (without a washout period) from omalizumab to mepolizumab, as in everyday clinical practice, was safe and effective. Interestingly, no additional efficacy was observed during the first half of the study when both omalizumab and mepolizumab were present in the patient, and there was no clinical worsening when omalizumab was washed out in the second half of the study. In one case [27], omalizumab was reintroduced because of a flare-up of allergy symptoms on mepolizumab therapy. In the study by Fox and Rotolo [29], the administration of two biologics was decided in agreement with the patient’s family, considering that the authors underlined no sensitivity to perennial allergens before starting omalizumab. In the study by Serajeddini et al. [22], dupilumab was added to anti-IL-5 mAB when elevated FeNO, airway smooth muscle dysfunction, and severe airway hyperresponsiveness (AHR), all signs of IL-4/IL-13 inflammation, were present, determining uncontrolled asthma symptoms.

The study by Baccelli et al. [24] provides the longest follow-up for patients who received a combination of biologics for this indication, which is over 3 years. Except for mild headaches in one study [29], no other side effects were reported by the authors.

The strongest evidence against combining biologics for the treatment of severe asthma derives from the only RCT performed, the itepekimab trial [21], which showed no beneficial effect compared to placebo in 74 patients treated with a combination of itepekimab and dupilumab. In the real-life studies included in this review, 2 patients failed to obtain complete symptoms control in the study by Serajeddini et al. [22], and 4 patients in the study by Lommatzsch et al. [34] discontinued combination therapy for severe asthma because of clinical ineffectiveness.

Cycling biologic therapy represents another strategy that has been used if a single biologic agent has not been able to achieve optimal control of asthma symptoms. One patient [32] who received treatment with dual biologic therapy (omalizumab plus mepolizumab and then benralizumab) achieved optimal control with cycling therapy with dupilumab and mepolizumab. According to the authors of this strategy, cycling therapy could also represent a “step down” treatment in patients who positively responded to dual biologic therapy [31].

Among the group of patients with severe asthma and insufficiently controlled typical T2-comorbidities, a dual biologic therapy was administered in the majority of cases for the treatment of CRSwNP, CSU, and AD [33,34,35,36,37,38,39,40,41], in 1 case for EoE [42], and in 3 cases for ABPA [43,44,45]. Many combinations of biologic agents have been described. Sensitivity to perennial allergens, BEC, values of FeNO, as well as clinical features (OCS dependence) have guided the clinicians in the decision and in the selection of a certain biological agent when there was concurrent severe asthma. Omalizumab can be given as add-on therapy for the treatment of chronic spontaneous urticaria in adult and adolescent patients with inadequate responses to H1 antihistamine treatment, and dupilumab is indicated for the treatment of moderate to severe atopic dermatitis, so both were prescribed following the indications of the regulatory authorities. The study by Pitlick and Pongdee [33], one of the largest so far, including 25 patients, showed no side effects deriving from combining two biologic agents, both for the treatment of severe asthma and a comorbidity and for the treatment of two different type 2 inflammatory diseases. In this group of patients who received a dual biologic therapy for the treatment of severe asthma and type 2 comorbidities, even though a lower number of patients were analyzed compared to the other group of severe asthma only, some degree of improvement was observed in almost all studies, and no adverse events have been reported. It is important to highlight that recently, asthma biologic agents have been approved for the treatment of other type 2 inflammatory diseases. In 2022, dupilumab was approved by the FDA for eosinophilic esophagitis since, in a Phase 3 RCT controlled with placebo [51], it was associated with histologic improvement and reduction of symptoms in 240 adults and adolescents. Future studies will evaluate the efficacy in patients with EoE and comorbid severe asthma (in the RCT, 90 patients, or 38% of the total, had asthma). Dupilumab is also being studied for the treatment of chronic spontaneous urticaria in patients who remain symptomatic despite the use of H1 antihistamines and who are naive to, intolerant of, or incomplete responders to omalizumab (LIBERTY-CSU CUPID, NCT04180488) [52]. The new monoclonal antibody tezepelumab, in addition to severe asthma, is currently being evaluated for other potential indications, including CSU (INCEPTION, phase 2b study, NCT04833855) [53], CRSwNP (WAYPOINT, phase 3 study, NCT04851964) [54], eosinophilic esophagitis (CROSSING, phase 3 study, NCT05583227) [55], and AD (in the phase 2b study, NCT03809663, tezepelumab as monotherapy did not reach the targeted efficacy level pre-established in subjects with moderate to severe AD) [56]. In October 2021, tezepelumab was granted orphan drug designation by the FDA for the treatment of eosinophilic esophagitis.

Biologics have been reported to be effective in the treatment of ABPA [57,58]. The recommended first-line treatments are oral corticosteroids and systemic therapy with azoles. Recently, biologics have been successfully added to treatment as they target cytokines involved in pathogenesis. Biologics might be used in patients with treatment-refractory ABPA, uncontrolled asthma despite oral corticosteroids, and in patients who experienced adverse effects with glucocorticoids and antifungal therapy or had contraindications to these therapies [59]. Future RCTs will explore the role of these agents in ABPA.

Increases in peripheral blood eosinophils may occur with dupilumab [60]. In fact, it inhibits IL-4 and IL-13-induced eosinophil migration from the blood to tissues [61]. It is a common event, not generally associated with clinical symptoms or impact on efficacy, and it is very often transient. The development of clinically relevant eosinophilia (>1500 cells/mL) is rare but can occur. Hypereosinophilia has in fact been observed in 4% to 25% of patients treated with dupilumab and persisted after 6 months in 14% of these patients. Furthermore, cases of eosinophilic granulomatosis with polyangiitis (EGPA), eosinophilic vasculitis, and eosinophilic pneumonia have been described [6]. Eger et al. [62] described 4 patients, all previously treated with an anti-IL5 biologic drug for OCS-dependent asthma, with poor lung function and high levels of FeNO, who developed eosinophilic complications after initiation of dupilumab, administered to try to obtain better control of the sinonasal disease. Despite an initial good response after switching, eosinophilic complications occurred. All 4 patients discontinued dupilumab and switched back to anti-IL-5 or anti-IL-5R. When dupilumab was interrupted, sinonasal symptoms relapsed in two patients, as reported by the authors. A combination therapy with anti-IL5/5R and anti-IL4/IL13R has been administered in the aforementioned real-life studies [46,47,48,49] in patients on therapy with dupilumab and experiencing reactive hypereosinophilia to provide better control of both asthma and nasal polyposis while preventing other hypereosinophilia-related disorders. In the cases described above, a combination of biologics was prescribed for the following reasons:-Eosinophilic vasculitis treated with the addition of anti-IL5 [46];-Diagnosis of EGPA after switching from anti-IL5 to dupilumab. Dupilumab was reintroduced for worsening respiratory function and increased FeNO levels [47];-Eosinophilic conjunctivitis, treated with the addition of an anti-IL5 [48];-Severe CRSwNP symptoms followed the discontinuation of dupilumab after the patient switched back to an anti-IL5 agent because of anti-IL4/IL13R-induced side effects (worsening respiratory symptoms and increased BEC) [49].

Dealing with hypereosinophilia may be challenging for clinicians. Recently, an algorithm has been proposed to avoid unnecessary drug discontinuation and, looking for complications, better monitor and treat those patients at risk for eosinophil organ involvement [63].

The administration of a combination of biologics, concurrently or sequentially, was decided by the authors of the included articles following inflammation biomarkers (i.e., blood eosinophils, FeNO, serum IgE levels) when severe asthma was uncontrolled on a biologic monotherapy and according to patient comorbidities when the patient was affected by severe asthma and another T2 condition eligible for biologic treatment. The patient with an uncontrolled allergic and eosinophilic asthma phenotype was the most commonly described profile of a patient with severe asthma eligible for combination of biologics. Regarding the presence of comorbidities, many are administered in combinations, in the majority of the described cases, for the treatment of CRSwNP, CSU, and AD. Patient selection remains a key factor in order to optimize treatment outcomes. When conventional treatment could not control symptoms, even with a maximal therapy, leading to side effects, a combination of biologics might be considered.

This review presents several limitations. We conducted systematic literature research and, according to inclusion and exclusion criteria, chose a narrative review to describe published data, given the small number of included studies and the low evidence-based results. Moreover, a reporting bias is also present, since cases with unsuccessful treatments may not have been published.

Therefore, we presented the results objectively, providing an up-to-date overview of the existing literature on combinations among biologic agents approved for severe asthma. We also summarized the treatment efficacy and related adverse effects.

## 5. Conclusions

There is a lack of data regarding combining biologics for the treatment of severe asthma and type 2 comorbidities. Several case reports and case series showed positive outcomes; however, no guidelines provide any recommendations on this use.

Further studies on a larger number of patients are required to assess the safety and efficacy of combination therapies for severe asthma patients and for those patients who cannot obtain control of both upper and lower airway disease with biologic monotherapy or suffer from type 2 immunity-driven diseases. Clinical trials in particular, with rigorous study design and methodology, should demonstrate whether the positive outcomes published so far in real-world studies are replicable. To date, the only RCT performed showed no beneficial effect when itepekimab and dupilumab were combined [21].

The high costs of combination biologic therapy (estimated at around USD 60,000 to USD 80,000 per year) [36] may represent a big deal, limiting the practicability of combining biologics in low-resource settings.

Anyway, uncontrolled severe asthma represents a great economic burden, considering the costs deriving from medications, multiple hospitalizations, and loss of productivity. Therefore, a step-up therapy with a combination of biologics might be considered since it could result in decreased hospitalizations and overall reduced healthcare costs.

Since few head-to-head studies have been conducted about switching biologics and there are no clear guidelines and predictors to prescribe the optimal biologic, a multidisciplinary approach is essential to find the most appropriate timing to consider the switch, the most effective biologic agent to switch to, or before deciding to start treatment with two biologics combined [35].

Concurrent treatment with two biologics has been taken into consideration as an add-on treatment for severe asthma in some patients who failed to obtain complete control of symptoms with an initial single biologic agent administered alone, when the presence of a T2 comorbidity required the use of another asthma-approved biologic, and in some cases of persistent reactive hypereosinophilia or eosinophilic disorders under treatment with dupilumab.

Figure 2 shows approved indications of asthma biologic agents and possible combination strategies, according to patient phenotype and type 2 comorbidities.

Since many RCTs are ongoing for biologics to treat several pathologies and the indications of asthma-approved biological agents are increasing, there might be multiple diseases that could be targeted by this approach. However, the qualitative weakness of the included studies makes it difficult to come to any conclusion on the efficacy and safety of combining two biologics. For this indication, more comparison studies are needed that evaluate the real impact of these combination strategies, intensifying the search for biomarkers capable of differentiating the different therapeutic approaches based on the treatment response. Future studies are needed in order to investigate the real opportunities for this strategy and enlarge our knowledge about persistent inflammation in severe asthma or in non-responder or partial responder patients who received an appropriate biologic drug, as well as intensify our basic research studies to further understand the inflammatory mechanisms or other pathways that act in these patients, not only at a systemic level but even at a local or tissue-specific level.

## Figures and Tables

**Figure 1 jpm-13-01594-f001:**
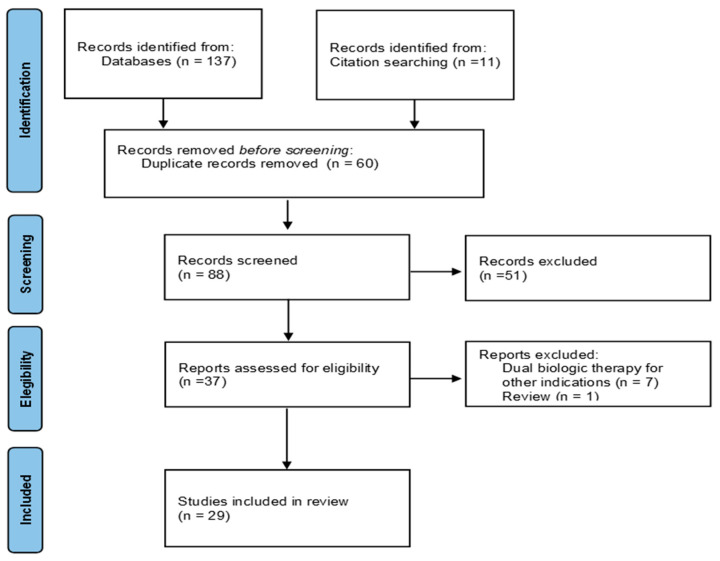
PRISMA 2020 Flowchart Diagram of the selected articles.

**Figure 2 jpm-13-01594-f002:**
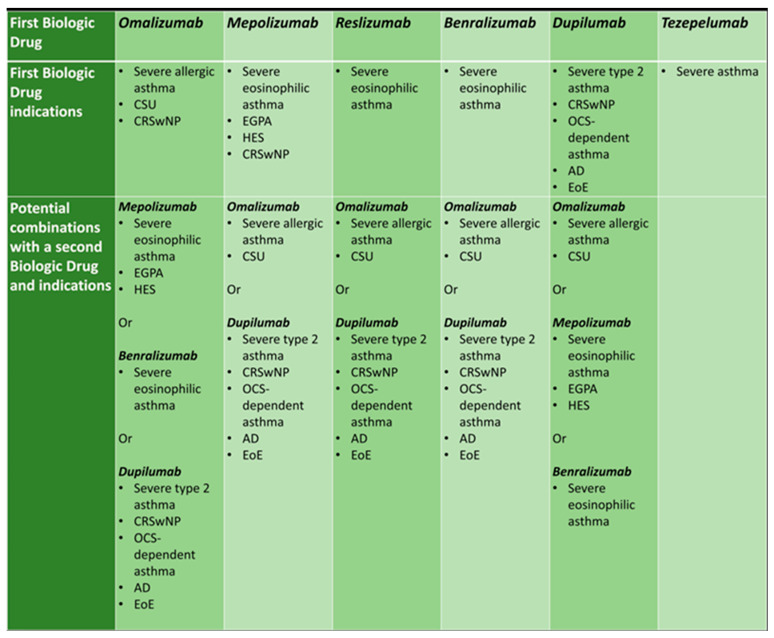
Approved indications of asthma biologic agents and possible combination strategies, according to patient phenotype and comorbidities.

**Table 1 jpm-13-01594-t001:** Articles examining the association of biological therapies in severe asthma.

Author(s), Year	Study Type	Study Population	Pathologies	Biologic Agents	Follow Up	Efficacy	Safety
(A) Summary of the characteristics of the studies with dual biologic therapy for severe asthma insufficiently controlled with a biologic monotherapy.							
Wechsler et al., 2021, [21]	Phase II Trial	n = 74	Severe asthma	DUP + ITE	12 we	27% of patients had an event indicating loss of asthma control, no increase in pre-bd FEV1, improved asthma control (ACQ and AQLQ)	70% had AE, nasopharyngitis, allergic rhinitis, nausea, and back pain the most common
Serajeddini et al., 2023, [22]	Case series	n = 8	Severe eosinophilic asthma (n = 8)	BEN + DUP (n = 4) MEP + DUP (n = 2) RES + DUP (n = 2)	Median: 10 mo	Significant improvement in clinical, functional, and inflammatory parameters (full data provided). 2 patients did not receive complete symptoms control	No AE reported
Thomes and Darveaux, 2018, [23] *abstract only*	Case series	n = 3	Severe allergic and eosinophilic asthma	OMA + MEP	Unknown	Reduced number of exacerbations, tapered daily OCS dose and provided a better control of symptoms.	Unknown
Baccelli et al., 2022, [24]	Case report	68 yo F	Severe allergic and eosinophilic asthma	OMA + MEP	3 y	Reduced BEC (200 vs. 2330 cells/mcl), increased exercise capacity (6MWT 280 vs. 160 m) and lung function (FEV1 1.32 vs. 1.08 L), improvement in QOL, reduced exacerbations, reduced daily OCS and SABA use.	No AE reported
Dedaj and Unsel, 2018, [25]	Case report	55 yo F	Severe allergic and eosinophilic asthma	OMA + MEP	6 mo	Reduced OCS daily dose, reduced exacerbations and no ER visit or hospitalization	No AE reported
Bergmann et al., 2022, [26]	Case report	53 yo M	Severe allergic and eosinophilic asthma	OMA + MEP OMA + BEN	17 mo >7 mo	OCS and LTOT discontinued, improvement of exercise capacity at 6MWT, improved QOL.	No AE reported
Domingo et al., 2020, [27]	Case report	55 yo F	Severe allergic and eosinophilic asthma	OMA + MEP	24 mo	Improvement in the FEV1 (96% pred. vs. 22% pred.), OCS dose reduction.	No side effects observed
Sezgin et al., 2023, [28]	Case report	52 yo M	Severe allergic asthma	OMA + MEP	Unknown	Improvement in the control of symptoms, improved QOL and increased respiratory function. OCS stopped.	No side effects observed
Fox and Rotolo, 2021, [29]	Case report	12 yo F	Severe allergic and eosinophilic asthma	OMA + MEP	24 mo	Weaned off OCS, improvement in QOL	Mild headache
Phan et al., 2018, [30] *abstract only*	Case report	16 yo F	Severe allergic asthma, allergic rhinitis	OMA + MEP	4 mo	Symptoms improved, higher QOL.	No AE observed
(B) Summary of the characteristics of the studies with cycling biologic therapy for severe asthma.							
Hamada et al., 2021, [31]	Case report	43 yo F	Severe eosinophilic asthma, ECRS, EOM	BEN + DUP as Cycling Therapy	11 mo	No new exacerbations, SABA and OCS not used, BEC = 0, Lund-Mackay score decreased to zero, reduction of FeNO levels	No AE reported
Hamada et al., 2021, [32]	Case report	47 yo M	Severe allergic and eosinophilic asthma, CRSwNP	MEP + DUP as Cycling Therapy	12 mo	No emergency department visits or hospital admissions, SABA and OCS not used, no decrease in ACT or elevation in eosinophil count, Lund-Mackay score decreased to zero	No adverse effects occurred
(C) Summary of the characteristics of the studies with dual biologic therapy for uncontrolled severe asthma and type 2 comorbidities							
Pitlick and Pongdee, 2022, [33]	Case series	n = 25	Severe asthma: 10 patients Severe asthma and CRSwNP: 1 patient Severe asthma and EGPA: 1 patient Severe asthma and CSU: 1 patient T2 comorbidities: 12 patients	OMA + MEP (n = 11) OMA + DUP (n = 6) OMA + BEN (n = 4) MEP + DUP (n = 3) OMA + DUP + MEP (n = 1)	1–60 mo Median: 17.5 mo	Unknown	No AE reported
Lommatzsch et al., 2022, [34]	Case series	n = 25 *	Severe asthma: 7 patients Severe asthma and AD: 2 patients Severe asthma and CSU: 1 patient Severe asthma and EGPA: 1 patient Severe asthma and CRSwNP: 4 patients	BEN + DUP (n = 5) DUP + MEP (n = 3) OMA + DUP (n = 2) OMA + MEP (n = 2) BEN + OMA (n = 1) MEP/BEN + OMA (n = 1) OMA + RES/MEP (n = 1)	3–38 mo Median: 9 mo	Improvement in ACT and FEV1, reduced or stopped OCS use. Interrupted in 4 patients for ineffectiveness	No AE reported
Otten et al., 2023, [35]	Retrospective observational study	n = 7	Severe eosinophilic asthma and CRSwNP: 7 patients	DUP + BEN (n = 4) DUP + MEP (n = 2) RES + DUP (n = 1)	Unknown	Unknown	Unknown
Ortega et al., 2019, [36]	Case series	n = 3	Severe allergic and eosinophilic asthma and multiple type 2 comorbidities AD, allergic rhinitis, ABPA: 1 patient CSU, allergic rhinitis, CRSwNP: 1 patient CRSwNP, ABPA, allergic conjunctivitis: 1 patient	OMA + DUP (n = 1) OMA + MEP/BEN (n = 2)	9 mo Pt.1 31 mo Pt.2 19 mo Pt.3	OCS stopped, better control of symptoms. Pt. 3 was hospitalized for an exacerbation and switched from OMA + BEN to DUP monotherapy.	No AE reported
Caskey and Kaufman, 2021, [37]	Case report	51 yo M	Severe eosinophilic asthma, CSU	MEP + OMA	4 y	Improvement in urticaria symptoms and QOL. Asthma controlled, no OCS required, no additional FEV1 improvement.	No AE reported
Volpato et al., 2020, [38]	Case report	60 yo M	Severe eosinophilic asthma, CSU	MEP + OMA	3 y	Stopped OCS use, improvement in FEV1 (86 vs. 73% pred), no skin itching or anaphylactic manifestation. No new hospitalizations, one mild exacerbation.	No AE reported
Nicolaides and Khan, 2019, [39] *abstract only*	Case report	50 yo F	Severe asthma, AERD, CRSwNP, urticaria	BEN + OMA	>2 mo	Improved control of both urticaria and asthma symptoms, reduced need for OCS.	No AE observed
Can Bostan et al., 2023, [40]	Case report	NA	Severe asthma, urticaria	OMA + MEP	6 mo	Both diseases controlled	No AE reported
Tongchinsub and Carr, 2017, [41] *abstract only*	Case report	56 yo F	Severe eosinophilic asthma, CRSwNP, eosinophilic mastoiditis	OMA + MEP	5 mo	Stopped OCS use.	No AE reported
Han and Lee, 2018, [42]	Case report	67 yo F	Severe asthma, EoE	MEP + OMA	Unknown	Improvement of asthma and GI symptoms, OCS stopped.	No AE reported
Laorden et al., 2022, [43]	Case series	n = 3	Severe asthma and ABPA: 3 patients	OMA + BEN (n = 2) OMA + MEP (n = 1)	2 y	Improvement of asthma symptoms, no exacerbations, improvement of lung function, reduction of OCS.	No AE reported
Curtiss et al., 2023, [44] *abstract only*	Case series	n = 2	Severe asthma and ABPA	DUP + MEP DUP + BEN	Unknown	Weaned daily OCS dose. Improvement of asthma symptoms and decreased exacerbation frequency. 1 mild exacerbation.	No AE reported
Altman et al., 2017, [45]	Case report	58 yo F	Severe asthma and ABPA	OMA + MEP	7 mo	Full return of ADLs, discontinuation of OCS and supplemental oxygen, reduced BEC (0 vs. 1100 cells/mcl) and IgE (298 vs. 1730 IU/mL).	No AE reported
(D) Summary of the characteristics of the studies with dual biologic therapy for dupilumab induced hypereosinophilia and eosinophils related disorders							
Descamps et al., 2021, [46]	Case report	61 yo F	Severe asthma, CRSwNP, dupilumab induced eosinophilic vasculitis	DUP + BEN	>16 mo	Complete and stable eosinophil depletion, skin lesions completely healed, no new vasculitic manifestations.	No AE reported
Anai et al., 2022, [47]	Case report	42 yo M	Severe asthma, EGPA	MEP + DUP	>16 we	Reduced symptoms, FEV1 improvement (4.43 vs. 3.84 L), reduced FeNO levels and BEC.	No AE reported
Philipenko et al., 2020, [48] *abstract only*	Case report	28 yo M	Severe eosinophilic asthma, AD, dupilumab induced conjunctivitis	DUP + MEP	Unknown	Reduction of BEC, improvement in ACQ, no asthma exacerbations, conjunctivitis resolved.	No AE reported
Briegel et al., 2021, [49]	Case report	24 yo F	EGPA, CRSwNP	DUP + BEN	1.5 y	Improvement of pulmonary and nasal symptoms, reduction of OCS daily dose.	No AE reported

* 10 patients received a dual biologic therapy with a biologic approved for severe asthma and a second biologic drug for another concomitant disease. ABPA—allergic bronchopulmonary aspergillosis; ACT—Asthma Control Test; ACQ—Asthma Control Questionnaire; AD—atopic dermatitis; ADLs—activities of daily living; AE—adverse event; AERD—aspirin exacerbated respiratory disease; AQLQ—Asthma Quality of Life Questionnaire; BEC—blood eosinophil count; BEN—benralizumab; CRSwNP—chronic rhinosinusitis with nasal polyposis; CSU—chronic spontaneous urticaria; DUP—dupilumab; ECRS—eosinophilic chronic rhinosinusitis; EGPA—eosinophilic granulomatosis with polyangiitis; EoE—eosinophilic esophagitis; EOM—eosinophilic otitis media; ER—emergency room; F—female; FeNO—fractional exhaled nitric oxide; FEV1—forced expiratory volume in 1 s; ITE—itepekimab; LTOT—long term oxygen therapy; M—male; MEP—mepolizumab; mo—months; NA—not available; OMA—omalizumab; OCS—oral corticosteroids; QOL—quality of life; RES—reslizumab; SABA—short acting β2 agonist; y—year; yo—year old; we—weeks; 6MWT—6 min walking test.

## Data Availability

Data are contained within the article.

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
