# Peer review of "Combination of Biological Therapy in Severe Asthma: Where We Are?"

_jpm, 2023, doi:10.3390/jpm13111594_

Round 1

Reviewer 1 Report

Comments and Suggestions for Authors

Please give citation for the database used for searching

In Study Selection of papers how many numbers of papers are in  exclusion criteria

Elucidate the data from the selected 29 articles in chart by using your creativity. 

The paper adequately highlights the lack of sufficient data on combining biologics for severe asthma and associated type 2 comorbidities. It appropriately references case reports, case series, and clinical trials, reinforcing the statements made.

Overall, the article effectively addresses the need for further research, economic considerations, and the importance of guidelines in the context of combining biologics for severe asthma and type 2 comorbidities.

Comments on the Quality of English Language

The language used is clear and concise, contributing to the overall readability of the article. However, minor grammatical edits may be beneficial for precision and fluency.

Author Response

First of all we want to thank the reviewers for taking the time to review our manuscript and their brilliant suggestions.

For these reasons, taking into account their comments we have modified and tried to improve the manuscript.

Point 1: Please give citation for the database used for searching

Answer: We really thank the reviewer if the indication of database used was not clear, so we changed the paragraph as follows: A systematic search was performed following the Preferred Reporting Items for Systematic Reviews and Meta-Analyses (PRISMA) guidelines to identify the current literature about dual therapy with asthma approved biologics for a combined treatment of severe asthma and for treating concurrent severe asthma and type 2 comorbidities. The search was conducted on the online databases MEDLINE (PubMed) from the National Library of Medicine (NLM) and Scopus, from inception to September 2023, and was followed by manual literature searches in the reference lists of the included articles to identify additional articles about this topic. The research string was as follows: (dual [Title] OR combination [Title] OR simultaneous [Title] OR combining [Title] OR combined [Title] OR concomitant [Title]) AND (biologic [Title] OR biologics [Title] OR biologic therapy [Title] OR therapy [Title] OR biologics therapy [Title] OR monoclonal antibody [Title] OR monoclonal antibodies [Title] OR biologic treatment [Title] OR targeting [Title]) AND (severe asthma [Title] OR uncontrolled asthma [Title] OR t2 comorbidities [Title] OR type 2 comorbidities [Title] OR asthma and CRSwNP [Title] OR asthma and atopic dermatitis [Title] OR asthma and urticaria [Title] OR asthma and EGPA [Title] OR asthma and eosinophilic esophagitis [Title] OR asthma and ABPA [Title] OR omalizumab [Title] OR IgE [Title] OR Anti IgE [Title] OR mepolizumab [Title] OR reslizumab [Title] OR IL5 [Title] OR anti-IL5 [Title] OR benralizumab [Title] OR IL5R [Title] OR anti-IL5R [Title] OR dupilumab [Title] OR IL4/IL13 [Title] OR anti-IL4/IL13 [Title] OR Tezepelumab [Title] OR TSLP [Title] OR anti TSLP [Title] OR omalizumab and mepolizumab [Title] OR omalizumab and benralizumab [Title] OR IgE and IL5 [Title] OR omalizumab and dupilumab [Title] OR mepolizumab and dupilumab [Title] OR benralizumab and dupilumab [Title] OR omalizumab and reslizumab [Title] OR reslizumab and dupilumab [Title] OR IgE and IL4/IL13 [Title] OR IL5 and IL4/IL13 [Title]).

Point 2: In study selection of paper how many numbers of papers are in the exclusion criteria.

Answer: We really thank the reviewer for his comment. We proposed this information in the M&M section as follows: “The initial literature search generated 137 potentially eligible articles from the aforementioned databases, plus 11 records identified additionally by manual search. A total of 60 duplicates were identified and removed. After excluding 59 articles (51 off-topic, 1 review and 7 about combining asthma approved biologic agents with other monoclonal antibodies for different therapeutic indications), only 29 articles were included in this review according to the prespecified inclusion and exclusion criteria. A flow chart showing the study selection is presented in Figure 1.”

Point 3: elucidate the data from the selected 29 articles in chart by using your creativity

Answer: We really thank the reviewer for his comment. We proposed this information in the Table 1A, 1B, 1C, 1D according to the major and more relevant feature of the paper selected: Author(s), year of publication, Study Type, Study Population, Pathologies, Biologic Agents, Follow up, Efficacy and Safety. We proposed this information through multiple tables, according to published data, with the aim of making easier the interpretation of data discussed in the manuscript. We really hope it resulted in a understandable way.

Point 4: The paper adequately highlights the lack of sufficient data on combining biologics for severe asthma and associated type 2 comorbidities. It appropriately references case reports, case series and clinical trials, reinforcing the statement made.

Answer: We would like to thank the Reviewer for his pleasant and stimulating comment. We have tried to systematically address a topic that certainly deserves greater future investigation because it underlies an unsatisfied health need.

Point 5: Overall, the article effectively addresses the need for further research, economic consideration and the important of guidelines in the context of combining biologics for severe asthma and type 2 comorbidities.

Answer: we thank again the Reviewer for his comments and suggestions. We have improved the manuscript about this points adding some reinforced statements in introduction and conclusions sections, respectively, as follows: "Limited preliminary data, mostly derived from observational studies and case series suggest this strategy may be safe in selected patients even if combining therapies is not permitted in some countries or by certain payors [Frix AN, Heaney LG, Dahlén B, Mihaltan F, Sergejeva S, Popović-Grle S, et al. Heterogeneity in the use of biologics for severe asthma in Europe: a SHARP ERS study. ERJ Open Res. 2022 Oct;8(4):00273–2022.] and before considering the opportunity to combine biological drugs a complete reassessment of patient at time of treatment failure or partial or deterioration of response can be recommended, which may include pulmonary function tests, airway inflammatory cells count, imaging and/or bronchoscopy for their complicating disorders and detecting neutralizing drug antibodies, reported in 1–4% of participants in clinical trials [Rogers L, Jesenak M, Bjermer L, Hanania NA, Seys SF, Diamant Z. Biologics in severe asthma: A pragmatic approach for choosing the right treatment for the right patient. Respir Med. 2023 Nov;218:107414.]."

"Since many RCTs are ongoing for biologics to treat several pathologies and the indications of asthma approved biological agents are increasing, there might be multiple diseases that could be targeted by this approach. However, the qualitative weakness of included studies make it difficult to come to any conclusion on the efficacy and safety of combining two biologics for this indication moreover are needed comparison studies that evaluate the real impact of these combination strategies intensifying the search for biomarkers capable of differentiating the different therapeutic approaches based on the treatment response. Future studies are needed in order to investigate the real opportunities for this strategy and enlarge our knowledge about persistent inflammation in severe asthma or in non-responder or partial responder patients who received an appropriate biologic drug, also intensifying our basic research studies to further understand the inflammatory mechanisms or other pathways that act in these patients, not only at a systemic level but even at a local or tissue specific level."

Point 6: minor grammatical edits may be benefical for precision and fluency

Answer: we provided a grammatical edit for the final version

Reviewer 2 Report

Comments and Suggestions for Authors

Minor comments

In this research article, the authors reviewed papers that discuss the use of combination therapy with biological drugs in the management of severe uncontrolled asthma and type 2 comorbidities. The manuscript is well-written, and the experimental design and data analysis are robust.

Point 1: What arе thе currеnt advancements in combining biological therapies for the treatment of severe asthma,   and how do these combinations impact thе managеmеnt of thе condition?

Point 2: Can wе identify specific biomarkers or patiеnt profilеs that would optimize thе sеlеction and sequencing of biological therapies in severe asthma,   and how doеs this influence trеatmеnt outcomes?

Point 3: What challеngеs and potеntial solutions еxist in thе implementation of combination biological therapies for severe asthma within hеalthcarе systеms,   and how can wе enhance patiеnt accеss and outcomеs in this evolving fiеld of treatment? 

Point 4: Thеrе wеrе no issues found with the study or thеir intеrprеtation.  Nonеthеlеss,  thеrе is a suggеstion for futurе research to delve deeper into thе underlying mechanisms,  which would providе valuablе insights.  

Good Luck

Comments on the Quality of English Language

Minor editing of English language required

Author Response

First of all we want to thank the reviewers for taking the time to review our manuscript and their brilliant suggestions.

For these reasons, taking into account their comments we have modified and tried to improve the manuscript.

Point 1: What arе thе currеnt advancements in combining biological therapies for the treatment of severe asthma,  and how do these combinations impact thе managеmеnt of thе condition?

Answer:  We thank the reviewer for this useful and stimulating comment. In the introduction we added a specific reference to the need to carefully re-evaluate the patient before considering any combination strategy between biologics. We have therefore modified the manuscript as follows: "Limited preliminary data, mostly derived from observational studies and case series suggest this strategy may be safe in selected patients even if combining therapies is not permitted in some countries or by certain payors [Frix AN, Heaney LG, Dahlén B, Mihaltan F, Sergejeva S, Popović-Grle S, et al. Heterogeneity in the use of biologics for severe asthma in Europe: a SHARP ERS study. ERJ Open Res. 2022 Oct;8(4):00273–2022.] and before considering the opportunity to combine biological drugs a complete reassessment of patient at time of treatment failure or partial or deterioration of response can be recommended, which may include pulmonary function tests, airway inflammatory cells count, imaging and/or bronchoscopy for their complicating disorders and detecting neutralizing drug antibodies, reported in 1–4% of participants in clinical trials [Rogers L, Jesenak M, Bjermer L, Hanania NA, Seys SF, Diamant Z. Biologics in severe asthma: A pragmatic approach for choosing the right treatment for the right patient. Respir Med. 2023 Nov;218:107414.]".

Point 2: Can wе identify specific biomarkers or patiеnt profilеs that would optimize thе sеlеction and sequencing of biological therapies in severe asthma,   and how doеs this influence trеatmеnt outcomes?

Answer:  We also thank the reviewer very much for this clarification. Limited to the biomarkers that we have available, we have not found any work at the moment that indicates which of these should be considered unequivocally predominant in the choice of a biological therapy. Rather, we observed how the overall assessment of the patient, which includes both the use of biomarkers and a careful clinical evaluation, provided the best direction in the therapeutic setting. We have therefore modified the manuscript in the discussion section as follows: "The administration of a combination of biologics, concurrently or sequentially, was decided by the authors of the included articles following inflammation biomarkers (i.e blood eosinophils, FeNO, serum IgE levels) when severe asthma was uncontrolled on a biologic monotherapy, and according to patient comorbidities when the patient was affected by severe asthma and another T2 condition eligible for biologic treatment. The patient with uncontrolled allergic and eosinophilic asthma phenotype was the most commonly described profile of patient with severe asthma eligible for combination of biologics. Regarding the presence of comorbidities, many are the administered combinations,, in the majority of the described cases for the treatment of CRSwNP, CSU and AD. Patient selection remains a key factor in order to optimize treatment outcomes. When conventional treatment could not control symptoms, even with a maximal therapy or leading to side effects, a combination of biologics might be considered."

Point 3: What challеngеs and potеntial solutions еxist in thе implementation of combination biological therapies for severe asthma within hеalthcarе systеms,   and how can wе enhance patiеnt accеss and outcomеs in this evolving fiеld of treatment?

Answer: We thank again the reviewer for this interesting insight. We decided to divide the problem into two parts: the former is of a pragmatic nature, referring to the current guidelines which do not recommend combination therapy of multiple biologics in severe asthma and to the non-reimbursability of similar strategies depending on the different payers; the latter is of clinical and diagnostic nature, referring to the lack of specific predictors of response to a therapy or clinical indicators. The solutions could certainly be found in conducting clinical trials or comparison studies that could evaluate the real impact of these combination strategies or by intensifying the search for biomarkers capable of differentiating the different therapeutic approaches based on the response. We have therefore modified the manuscript in the conclusions section as follows: "Since many RCTs are ongoing for biologics to treat several pathologies and the indications of asthma approved biological agents are increasing, there might be multiple diseases that could be targeted by this approach. However, the qualitative weakness of included studies make it difficult to come to any conclusion on the efficacy and safety of combining two biologics for this indication moreover are needed comparison studies that evaluate the real impact of these combination strategies intensifying the search for biomarkers capable of differentiating the different therapeutic approaches based on the treatment response."

Point 4 Thеrе wеrе no issues found with the study or thеir intеrprеtation.  Nonеthеlеss,  thеrе is a suggеstion for futurе research to delve deeper into thе underlying mechanisms,  which would providе valuablе insights. 

Answer: We truly thank the reviewer for these valuable and important suggestions that we hope we were able to grasp in order to improve our manuscript. Precisely because of these open questions, we underlined in the conclusions the need not only to compare the differences in therapeutic strategies but also to intensify our basic research studies to further understand the inflammatory mechanisms that intervene in these patients, not only at a systemic level but even at a local or tissue specific level. We have therefore modified the manuscript in the conclusions section as follows: "Future studies are needed in order to investigate the real opportunities for this strategy and enlarge our knowledge about persistent inflammation in severe asthma or in non-responder or partial responder patients who received an appropriate biologic drug, also intensifying our basic research studies to further understand the inflammatory mechanisms or other pathways that act in these patients, not only at a systemic level but even at a local or tissue specific level."

Minor editing of English language required

Answer: we provided a grammatical edit for the final version